# Afatinib for the Treatment of NSCLC with Uncommon EGFR Mutations: A Narrative Review

**Yingying Jiang [1], Xiaoxu Fang [1], Yan Xiang [1], Tingwen Fang [1], Jingwen Liu [1] and Kaihua Lu [2],***

[1] Department of Oncology, Nanjing Medical University, Nanjing 210029, China; jyy0324@stu.njmu.edu.cn (Y.J.); fang_xu@126.com (X.F.); xiangyan@stu.njmu.edu.cn (Y.X.); 13218000671@163.com (T.F.); nyliujingwen@163.com (J.L.)

[2] Department of Oncology, The First Affiliated Hospital of Nanjing Medical University, Nanjing 210029, China

*  Correspondence: lukaihua@njmu.edu.cn; Tel.: +86-136-0517-9453

**Abstract:** Afatinib, the world's first irreversible ErbB family (containing four different cancer cell epidermal growth factor receptors, including EGFR, HER2, ErbB3, and ErbB4) inhibitor, is a second-generation oral epidermal growth factor receptor tyrosine kinase inhibitor (EGFR-TKI). It can be used as a first-line treatment for locally advanced or metastatic non-small-cell lung cancer (NSCLC) with an EGFR-sensitive mutation or for patients with locally advanced or metastatic squamous lung cancer whose disease progresses during or after platinum-containing chemotherapy. Currently, with the use of third-generation EGFR-TKIs, afatinib is no longer clinically indicated as the first choice for patients with NSCLC who have EGFR-sensitive mutations. However, afatinib showed a considerable inhibitory effect in NSCLC patients with uncommon EGFR mutations (G719X, S768I, and L861Q) according to a combined post hoc analysis of the LUX-Lung2/3/6 trials. With the development of genetic testing technology, the detection rate of uncommon EGFR mutations is increasing. The aim of this paper is to describe in detail the sensitivity of rare EGFR mutations to afatinib and to provide information and a reference for those suffering from advanced NSCLC who have uncommon EGFR mutations.

**Keywords:** afatinib; epidermal growth factor receptor; uncommon mutations; rare mutations; non-small-cell lung cancer





## 1. Introduction

In current history, great strides have been made in the treatment of advanced non-small-cell lung cancer (NSCLC), and there is a consensus that epidermal growth factor receptor tyrosine kinase inhibitor (EGFR-TKI) therapy is preferred for patients who are EGFR-positive. In non-Asian groups, EGFR mutations account for 10% to 15% of NSCLC cases, while the incidence of such mutations rises to 40% in Asian populations. The majority of EGFR mutations are exon 19 deletion mutations and exon 21 L858R point mutations, collectively referred to as common EGFR mutations, which are responsible for 80% to 90% of the total EGFR mutations observed. Patients carrying common EGFR mutations are the most beneficial population for EGFR-TKI molecular targeted therapy. The remaining approximately 10% of EGFR mutations are considered rare [1]. These variations contain exon 20 insertions and/or point mutations (such as S768I), compound mutations (such as S768I+G719X), uncommon mutations in exon 19, and less frequent exon 21 mutations (such as L861Q) [2,3]. Among them are the missense mutations G719X, S768I, and L861Q, respectively, which are common mutations among uncommon mutations and are known as significant rare mutations [4]. It is noteworthy that, unlike common mutations, these uncommon mutations exhibit varying degrees of sensitivity to EGFR-TKIs.

In a retrospective investigation conducted in China, the clinical traits and therapeutic outcomes of individuals with NSCLC with common and uncommon EGFR mutations were

evaluated. According to the study, patients with uncommon mutations who received first-line platinum-based chemotherapy or first-generation EGFR-TKIs seemed to have a lower objective response rate (ORR) and median progression-free survival (PFS) than clients with common mutations (ORR: 23.3% vs. 51.8%, $p = 0.003$; median PFS: 7.1 vs. 10.9 months, $p = 0.001$) [5]. Gefitinib and erlotinib, first-generation EGFR-TKIs, were investigated for their efficacy for the treatment of lung adenocarcinomas with the G719X, L861Q, and S768I mutations in a multicenter study from Taiwan. The outcomes were compared to those of patients with classical EGFR mutations. The results suggested that after receiving first-generation EGFR-TKIs, compared to patients with classical mutations, individuals with rare mutations had a significantly poorer ORR (41.6% vs. 66.5%, $p < 0.001$), median PFS (7.7 vs. 11.4 months, $p < 0.001$), and median overall survival (OS) (17.2 vs. 27.8 months, $p < 0.001$) [6]. A comprehensive worldwide retrospective study called UpSwinG was conducted to assess the effectiveness of NSCLC patients with uncommon EGFR gene mutations. Afatinib patients had a median time to treatment failure (TTF) of 11.3 months (95% CI: 8.5–14.9), which was higher than the 8.8-month (95% CI: 6.4–10.7) TTF shown in patients undergoing first-generation EGFR-TKIs [7]. According to the research mentioned above, chemotherapy or first-generation EGFR-TKIs are less profitable for treating NSCLC patients with uncommon EGFR mutations than they are for treating patients with common EGFR mutations. Compared to first-generation EGFR-TKIs and chemotherapy, afatinib may be a superior option for patients with uncommon EGFR mutations.

In 2018, the Food and Drug Administration (FDA) of the United States authorized afatinib as a first-line therapy for individuals with the uncommon EGFR mutations G719X/L861Q/S768I. In light of this, the current paper aims to offer a thorough analysis of the effectiveness of afatinib in the therapy of NSCLC patients with various uncommon EGFR mutations, which can help medical professionals decide what the most effective strategy for individuals with such mutations is.

## 2. Anti-Tumor Mechanism

Members of the ErbB family include the human epidermal growth factor receptor 2 (HER2), the epidermal growth factor receptor (EGFR), ErbB3, and ErbB4, which are all potently inhibited by the medication afatinib. As an irreversible blocker, it binds covalently to the tyrosine kinase domain of these receptors, leading to their inactivation and preventing the downstream signaling pathways involved in cell growth and survival [8–11] (Figure 1). Afatinib's distinctive anticancer therapeutic benefit—its irreversible binding to ErbB family receptors—offers higher potential to suppress the proliferation of a wide class of tumor cells and greater efficacy than the first-generation EGFR-TKIs [12]. Afatinib works in NSCLC with common EGFR mutations through this pharmacological mechanism, and some individuals with rare EGFR mutations are also susceptible to it.

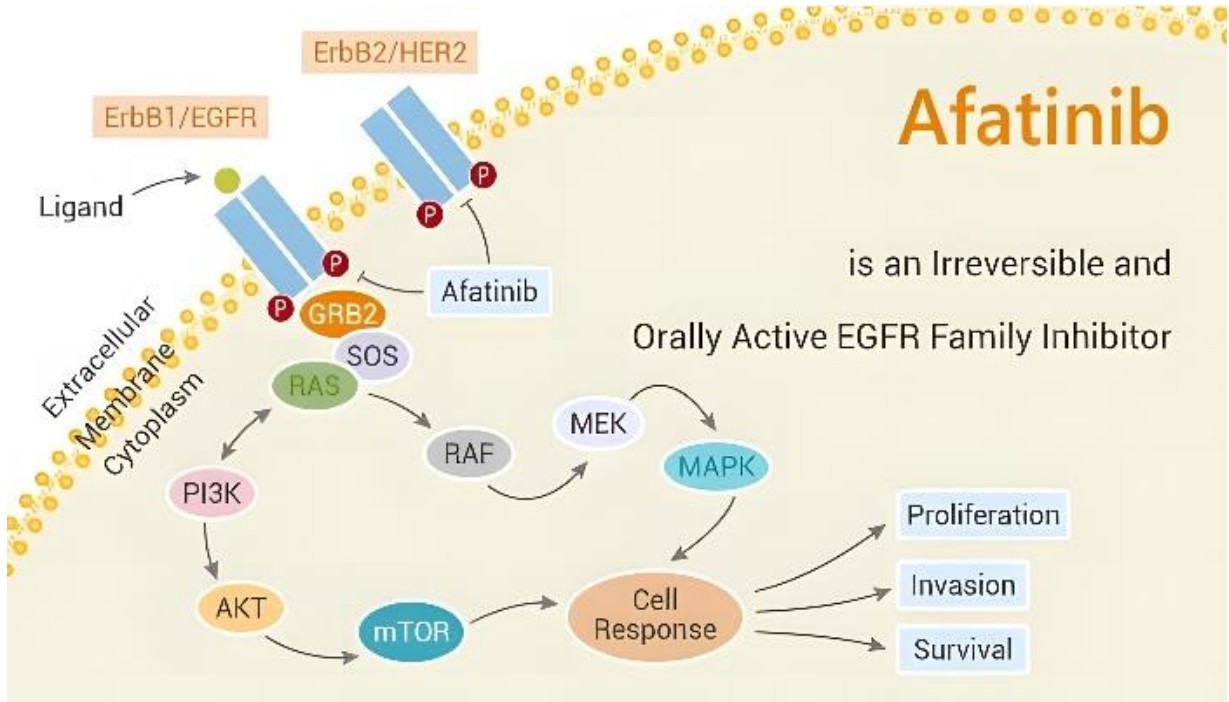

**Figure 1.** Anti-tumor mechanism of afatinib (picture from the Network of Cancer Research).

### 3. Afatinib's Effectiveness in Curing Various Uncommon EGFR Mutations

*3.1. Major Uncommon Mutations (G719X/S768I/L861Q)*

To understand the efficacy of afatinib for patients with advanced NSCLC carrying uncommon EGFR mutations, a pooled analysis of phase two trials (LUX-Lung2) and randomized phase three trials (LUX-Lung3 and LUX-Lung6) showed that individuals with major uncommon mutations, such as G719X, L861Q, or S768I, had substantially longer medians for PFS and OS than those with other uncommon EGFR mutations. Individuals who carried the G719X, L861Q, and S768I mutations had ORRs of 78%, 56%, and 100%, respectively. The results were 13.8 months (95% CI: 6.8–NE), 8.2 months (95% CI: 4.5–16.6), and 14.7 months (95% CI: 2.6–NE), respectively, for the median PFS values [13]. Therefore, afatinib appears to be more effective for patients with NSCLC carrying significant, rare EGFR mutations. Afatinib's potency in handling NSCLC with uncommon EGFR mutations was examined in another investigation. Patients with major uncommon mutations (G719X, L861Q, and S768I) with first-line therapy had a median TTF of 10.8 months (95% CI: 8.1–16.6), which was longer than patients with other rare mutations, including exon 20 insertion mutations (median TTF = 4.2 months), T790M (median TTF = 4.7 months), and more infrequent uncommon mutations (median TTF = 4.5 months). The median TTFs for patients with the G719X, L861Q, and S768I mutations were 14.7, 10.0, and 15.6 months, respectively, with ORRs of 63.4%, 59.6%, and 62.5% [14]. This investigation was followed by further expansion of the database sample size, and afatinib may be the initial treatment of preference for NSCLC patients with major uncommon EGFR mutations according to the findings, which showed that it was responsive to these mutations (median TTF: 12.6 months; 95% CI: 11.5–15.9; ORR: 59.0%) [15]. In addition, 90 individuals suffering adenocarcinomas with the G719X/S768I/L861Q mutation were included in a trial in Korea that used afatinib as the first-line treatment, with an ORR of 63.3%, a median PFS of 17.3 months (95% CI: 12.07–22.53), and a median OS of 28.5 months (95% CI: 20.22–36.77) [16]. Afatinib is effective for treating NSCLC patients who have significant, rare EGFR mutations, as shown in each of the aforementioned trials.

### 3.2. Exon 18 Mutations (Except G719X)

Around 3–5% of all EGFR mutations are triggered by exon 18 mutations, which, in addition to the most common G719X mutation, are also implicated in point mutation E709X, the more uncommon insertional deletion mutation E709_T710delinsX, and other molecular isoforms. The usefulness of EGFR-TKIs in NSCLC with EGFR E709X mutations has received less research [17] (Table 1).

**Table 1.** Afatinib's efficiency in treating NSCLC patients who had mutations in exon 18.

| Study | *n* | Mutation | Tumor Staging | Therapy Line | Response | PFS (mos) (95% CI) | OS (mos) (95% CI) |
|---|---|---|---|---|---|---|---|
| Hao Y., 2022 [18] | 11 | E709A/G/K, E709-T710delinsD | IV | first or above | N/A | 13.5 | 17.1 |
| Iwamoto Y., 2019 [19] | 1 | E709-T710delinsD | IV | sixth | PR | 7 | N.R. |
| Ibrahim U., 2017 [20] | 1 | E709-T710delinsD | IV | first | PR | N.R. | N.R. |
| An N., 2019 [21] | 1 | E709-T710delinsD | IIIA | second | PR | 11 | >21 |
| Wei Y., 2021 [22] | 1 | E709-T710delinsD | IIB | first | PR | 23 | N.R. |

Legend: NSCLC—non-small-cell lung cancer; N.R.—not reported; N/A—not applicable; PR—partial response; PFS—progression-free survival; OS—overall survival; mos—months.

A study with a small sample size of 15 NSCLC patients with exon 18 E709X or E709 T710delinsX mutations, all of whom received treatment with afatinib (*n* = 11) or third-generation EGFR-TKIs (*n* = 4), failed to find a statistically significant distinction between the two when it came to the median PFS (13.5 vs. 10.9 months, *p* = 0.774). The study found that individuals with NSCLC who had these particular mutations might profit from afatinib or third-generation EGFR-TKI therapy, despite the study's small number of participants [18]. The E709_T710delinsD mutation is considered the most commonly occurring deletion mutation in exon 18, and, although rare, isolated case reports imply that afatinib could be a useful therapeutic choice. A 56-year-old female patient diagnosed with advanced NSCLC was found to have the E709_T710insD mutation via next-generation sequencing (NGS). After receiving afatinib as sixth-line therapy, she achieved partial remission (PR) and experienced a progression-free survival of 7 months [19]. Several additional case reports also support the notion that afatinib is effective for patients carrying the E709_T710delinsD mutation [20,21]. Another patient with inoperable stage IIB lung cancer and the same E709_T710delinsD mutation achieved up to 23 months of PFS with first-line afatinib treatment. After experiencing disease progression, the patient received almonertinib and demonstrated stable disease [22]. This finding suggests that afatinib followed by sequential almonertinib treatment may serve as a viable therapy for NSCLC patients with the EGFR E709_T710delinsD mutation.

### 3.3. Uncommon Mutations in Exon 19

In addition to the typical 19 deletion variants, exon 19 also includes unusual deletion mutations, insertion mutations, and point mutations, and clinical data on afatinib treatment of related mutations are lacking (Table 2). Relevant in vitro experiments have shown evidence that afatinib is helpful against rare exon 19 mutations [23]. A case report on a patient with advanced NSCLC carrying the I740_K745dup mutation revealed that treatment with a first-generation EGFR-TKI achieved 4 months of PFS in the first line, while second-line treatment with afatinib extended the PFS up to 13.4 months, revealing that for individuals with such a mutation, afatinib would be a superior therapy option [24]. Afatinib showed a median PFS of 11.97 months in a Taiwanese study on 5 patients suffering from stage IV lung adenocarcinoma and who had the EGFR L747P or L747S mutation, particularly in comparison to only 0.92 months with a first-generation EGFR-TKI (*p* = 0.012), further illustrating the drug's ability to treat these uncommon mutations [25]. Additionally, a patient with the E746_L747delinsIP mutation achieved partial remission (PR) and 7 months of PFS after receiving afatinib as fourth-line therapy [26]. Another case report showed

that a patient with recurrent IV postoperative lung adenocarcinoma progressed again with multiple lines of treatment, including gefitinib, radiotherapy, and immunotherapy, and genetic testing was performed suggesting EGFR L858R and L747V mutations, and a 1-year PFS was obtained after treatment with afatinib [27]. There was also a case report of a patient with stage III lung adenocarcinoma who developed primary-lesion progression and brain metastasis after first-line concurrent radiotherapy and immunotherapy, with genetic testing suggesting an EGFR T751_I759delinsS mutation and an EGFR amplification; this was followed by afatinib treatment, with shrinkage of both brain and chest lesions and 9 months of PFS [28]. Collectively, these rare NSCLC cases carrying EGFR exon 19 insertions and rare deletion mutations suggest that some patients with these mutations may have favorable and durable responses to afatinib, supplying important clinical proof that afatinib treatment should be chosen for this particular group of patients with mutations.

**Table 2.** Afatinib's efficiency in treating NSCLC patients who had uncommon mutations in exon 19.

| Study | *n* | Mutation | Tumor Staging | Therapy Line | ORR (%) | Response | PFS (mos) (95% CI) | OS (mos) (95% CI) |
|---|---|---|---|---|---|---|---|---|
| Liang S.K., 2019 [25] | 5 | L747S or L747P | IV | first | 80 | N/A | 11.97 (6.37–17.57) | 16.62 (3.05–30.19) |
| Shan B.B., 2022 [24] | 1 | I740_K745dup | IV | second | 0 | SD | 13.4 | 21.3 |
| Zhang L., 2022 [26] | 1 | E746_L747delinsIP | IV | fourth | 100 | PR | >6.5 | N.R. |
| Kanbe M., 2022 [27] | 1 | L747V and 21 L858R | IV | sixth | 100 | PR | 12 | N.R. |
| Fang Y.F., 2021 [28] | 1 | T751_I759delinsS | IV | second | 100 | PR | 9 | N.R. |

Legend: NSCLC—non-small-cell lung cancer; N.R.—not reported; N/A—not applicable; PR—partial response; SD—stable disease; ORR—objective response rate; PFS—progression-free survival; OS—overall survival; mos—months.

### 3.4. Exon 20 Insertions and T790M Mutations

EGFR 20 insertion mutations are the third most common subtype of EGFR mutations and are seen in approximately 4–10% of EGFR-mutant NSCLC [29]. More than 60 unique EGFR 20 exon insertion mutations have been identified to date [30]. In a trial from China, 165 people suffering from advanced NSCLC who had EGFR exon 20 insertional mutations were included, and a total of 39 molecular variant types were identified, with the V796_D770insASV and D770_N771insSVD being the most prevalent mutations observed [31].

A total of 42 patients with well-informed exon 20 insertions were recognized in the aforementioned pooled analysis of afatinib treatment for NSCLC patients with rare mutations, with an overall ORR of 33% and a median TTF of 9.1 months (95% CI: 7.4–14.2). Patients among them with insertional mutations in A763, M766, N771, and V769 displayed afatinib sensitivity with a median TTF of 8.0 to 39.0 months and an ORR of 50% to 100%; notably, four patients with the A763 mutation obtained a median TTF of up to 39 months (95% CI: 8.2–39.0) [15] (Table 3). Afatinib was utilized to treat nine patients with NSCLC who had already had EGFR 20 insertional mutations in a large retrospective study: four with S768_D770dup, one with N771_H773dup, two with H773_V774dup, one with H773dup, and one with V769_D770insGVV. Only one patient with V769_D770insGVV had PR, and the DCR of all nine patients was 55.6%. Unfortunately, the PFS of patients with an EGFR 20 insertional mutation treated with afatinib was not described in the study [32]. In addition, there were several case reports of afatinib treatment of 20 insertional mutations. Notably, in one case, a patient with the H773dup mutation experienced a durable PFS of 4.5 years with afatinib treatment, suggesting that afatinib can provide long-lasting benefit to this rare patient with exon 20 insertion [33]. Another report was about a patient with advanced lung adenocarcinoma carrying the N771delinsKG mutation who achieved up to 10 months of PFS with afatinib [34]. An A767_S768insSVA-positive stage IV NSCLC patient treated with afatinib in the third line for up to three years without disease progression was also documented in a case study [35]. One stage IV NSCLC patient with the H773_V774insNPH mutation had afatinib as second-line therapy, and the patient's illness was in remission, and the PFS was greater than three years [36]. Other related cases reported responsiveness to afatinib with 20 insertion mutations, such as A767delinsASVD and D770_N771insSVD [37,38]. According to the findings presented above, certain people suffering from advanced NSCLC with EGFR

20 insertional mutations may benefit from afatinib medication in the first line and further on, and disease control can even last for more than 3 years. Furthermore, several studies have explored the combination of afatinib and cetuximab as a potential treatment option for NSCLC with EGFR exon 20 insertion mutations, suggesting that cetuximab may be able to increase the efficacy of afatinib in NSCLC with EGFR exon 20 insertion mutations [39,40].

**Table 3.** Afatinib's efficiency in treating NSCLC patients who had fully defined exon 20 insertion mutations.

| Study | *n* | Mutation | ORR (%) | TTF (mos) (95% CI) |
|---|---|---|---|---|
| Yang J.C., 2022 [15] | 42 | Overall | 33 | 9.1 (7.4–14.2) |
| | 4 | A763_Y764insFQEA; A763_V765dup | 50 | 39.0 (8.2–39.0) |
| | 5 | A767_S768insSVA; _V769dup/ASV; insASVD | 0 | 3.7 (1.0–36.0) |
| | 3 | D770_N771insGL/SVD | 0 | 3.8 (3.0–20.1) |
| | 9 | H773_R776insYNPY; _V774dup/insH; dup | 0 | 24.0 (6.1–NE) |
| | 2 | M766delinsMATL; insASV | 100 | 12.9 (11.6–14.2) |
| | 7 | N771_H773dup; _772insPHGH; delinsKG; _P772insGY | 71 | 10.0 (5.2–NE) |
| | 5 | S768_D770dup | 0 | 8.5 (NE–NE) |
| | 7 | V769_770INSV; _D770insASV/GVV | 75 | 8.0 (1.2–14.3) |

Legend: NSCLC—non-small-cell lung cancer; ORR—objective response rate; TTF—time to treatment failure; mos—months.

All of these trials highlight the variability in clinical responses to afatinib in EGFR20 insertional mutant NSCLC, with certain subtypes displaying a lack of response while others exhibit sensitivity and long-term efficacy (Table 4). This underscores the clinical significance of carefully selecting the appropriate treatment modality based on the specific types of 20 insertional mutations present. Extensive future evaluation of EGFR exon 20 insertions is needed to provide a basic database to help tailor treatment modalities for cancers with insertional mutations within EGFR exon 20, allowing more appropriate patients with EGFR 20 insertional mutations to benefit from afatinib therapy.

**Table 4.** Afatinib's efficiency in treating NSCLC patients who had exon 20 insertion mutations.

| Study | *n* | Mutation | Tumor Staging | Therapy Line | ORR (%) | Response | PFS (mos) (95% CI) |
|---|---|---|---|---|---|---|---|
| Zochbauer-Muller S., 2020 [33] | 1 | H773dup | N.R. | first | 0 | SD | >54 |
| Lin L., 2020 [34] | 1 | N771delinsKG | IV | first | 100 | PR | 10 |
| Chan R.T., 2018 [35] | 1 | A767_S768insSVA | IV | third | 0 | SD | 36 |
| Cai Y., 2019 [37] | 1 | A767delinsASVD | IV | second | 0 | SD | 7.4 |
| Urban L., 2021 [36] | 1 | H773_V774insNPH | IV | second | 100 | PR | >36 |
| Oyamada Y., 2021 [38] | 1 | D770_N771insSVD | IV | fourth | 100 | PR | N/A |
| van Veggel B., 2018 (with cetuximab) [39] | 4 | Overall | IV | second/third | 75 | N/A | 5.4 (0.0–14.2) |
| | 1 | S768_A770dup | | second | 100 | PR | 17.6 |
| | 1 | A771_H773dup | | second | 100 | PR | 4.4 |
| | 1 | H773dup | | third | 0 | SD | 2.7 |
| | 1 | A767_V769dup | | third | 100 | PR | >6.4 |

Legend: NSCLC—non-small-cell lung cancer; ORR—objective response rate; PFS—progression-free survival failure; N/A—not applicable; PR—partial response; SD—stable disease; mos—months.

T790M made up 20.3% of all patients in this pooled analysis of more than 1000 non-small-cell lung cancer individuals treated with afatinib, and it ranked second only to 20 insertions and major uncommon mutations, which were primarily seen in patients who had already been initially offered EGFR-TKIs, in frequency. Without undergoing EGFR-TKI medication, the ORR for individuals with T790M mutations was 26.2%, with a median TTF of 4.7 months (95% CI: 2.8–6.5). Patients with T790M mutations who had already previously taken EGFR-TKI treatment had an ORR of 17.8%, with a median TTF of just 4.0 months (95% CI: 3.5–5.6) [15]. In addition, a study from Taiwan, China, enrolled

44 NSCLC patients with primary T790M who were treated with EGFR0-TKIs and suggested that patients treated with afatinib and osimertinib exhibited longer PFS and OS than those who received gefitinib and erlotinib, with individuals receiving afatinib having a median PFS of 21.4 months (95% CI: 3.5–39.3 months) and a median OS of up to 22.4 months (95% CI: 1.7–43.2 months) [41]. Afatinib may be helpful for the treatment of initial T790M mutations and subsequent T790M resistant mutations, and its efficacy is superior to that of first-generation EGFR-TKIs. Some patients may also benefit long-term.

T790M is frequently regarded as a resistance mutation in EGFR-TKIs [42], and one study concluded that afatinib was less beneficial than third-generation osimertinib in terms of the survival of patients with NSCLC who obtained a T790M resistance mutation after failing first-line EGFR-TKI medication [43]. After receiving chemotherapy and erlotinib treatment, one patient with terminal lung cancer continued to deteriorate, and gene profiling revealed the T790M drug resistance mutation. Following the third-line treatment of afatinib and cetuximab, the disease established a partial remission, and it was still under control for 8 months [44]. It is predicted that the combination of afatinib and cetuximab may provide a novel treatment option for people with advanced NSCLC who have a T790M mutation following progress with a first-generation EGFR-TKI [45]. Based on the research mentioned above, afatinib is less effective against T790M resistance mutations when used alone, but patients suffering from this mutation may benefit from afatinib in combination with cetuximab therapy.

*3.5. Uncommon Exon 21 Mutations*

Among the rare mutations in exon 21, except the L861Q mutation mentioned above, other rare mutations in exon 21 have been rarely reported. In one case report, a patient with stage IV non-small-cell lung cancer who had the complicated uncommon mutation L833V/H835L in exon 21 experienced a significant decrease in primary and intrapulmonary metastases after receiving oral afatinib, and the patient had a PFS of 10 months by the time of the last follow-up [46]. Additionally, a patient with stage IV NSCLC with multiple metastases was found to have three uncommon EGFR mutations: EGFR R670w in exon 17, as well as H833V and H835L mutations in exon 21. After receiving afatinib as third-line therapy, this patient's chest lesions significantly decreased, and she had a 7-month PFS [47]. These two case reports imply that the use of afatinib could prove advantageous to individuals who have L833V or H835L mutations.

*3.6. Compound Mutations*

In this paper, compound mutations are defined as cases with two or more EGFR mutations as well as at least one uncommon mutation. The major rare mutations (G719X, L861Q, and S768I), compound mutations, exon 20 insertional mutations, and other variants were all split into four distinct cohorts in a Taiwanese trial of patients with uncommon mutations in NSCLC who were treated with EGFR-TKIs in the first line. In comparison to patients treated with the first-generation TKIs gefitinib and erlotinib, individuals with compound mutations managed by afatinib showed a tendency toward longer PFS (12.1 vs. 8.8 months, $p = 0.074$) and OS (25.7 vs. 15.8 months, $p = 0.052$) [48].

Individuals with compound mutations getting afatinib therapy experienced a median TTF of 12.6 months and a median OS of 23.4 months in accordance with the UpSwinG research, compared to a median TTF of 14.3 months and a median OS of 24.5 months in the main uncommon group (G719X, L861Q, and S768I) [7]. Complex mutations and significant uncommon mutations and minor uncommon mutations (exon 20 insertion, S768I, and de novo T790M) were divided into three groups in a study involving multiple centers in Korea and carried out on NSCLC patients with uncommon EGFR mutations. The research showed a median time on treatment (TOT) of 20.3 months (95% CI: 15.1–25.5) and a median OS of 30.6 months (95% CI: 26.3–34.8) in the major rare mutation group treated with afatinib. On the other hand, the compound mutation group exhibited a median TOT of 12.3 months (95% CI: 7.7–17.0) and a median OS of 29.1 months (95% CI: 20.4–37.7) [49]. Collectively,

both of these studies suggest that major uncommon and compound mutation categories among uncommon EGFR mutations are more responsive to afatinib therapy.

The previously mentioned pooled analyses revealed an ORR of 63.9% and a median TTF of 11.5 months (95% CI: 9.5–13.8 months; n = 182) among patients with NSCLC who had a compound mutation and were given afatinib as the initial therapy. The longest median TTF was 16 months (95% CI: 14.2–20.5 months; n = 90) in the combined major uncommon mutation group, but only 12.5 months (95% CI: 3.8–13.1 months, n = 11) for those with combined 20 insertional mutations, 4.7 months (95% CI: 3.0–6.5 months, n = 11) for those with combined T790M, and 11.5 months for those with other rare mutations (95% CI: 7.0–12 months, n = 88). The ORR for those with compound mutations in second-line and beyond was 21.8% with afatinib, with a median TTF of only 4.4 months [15], indicating that the first-line use of afatinib is especially effective for patients carrying rare compound mutations, particularly those with combined major uncommon mutations.

*3.7. Rare Mutations following Resistance to Osimertinib*

Several in vitro trials on osimertinib resistance have shown that some patients who are resistant to osimertinib develop G724S or L718Q/V mutations that limit the activity of osimertinib and that afatinib regresses tumors carrying these mutations [50–53]. Based on the results of studies related to the osimertinib resistance gene, several case reports have involved afatinib treatment-related resistance mutations.

One case showed a patient with stage IV lung cancer that progressed after second-line treatment with osimertinib, with resistance gene testing suggesting an EGFR L718V mutation, who was treated with afatinib in the third line with significant reduction in mediastinal lymph node metastases after 1 month and subsequent brain metastases in the patient's treatment, which included local brain radiotherapy and one cycle of bevacizumab to prevent brain edema, with the results showing that afatinib enabled the patient to achieve a 6-month PFS, except for the central nervous system, as of the last follow-up [54]. In another case report, a patient who had received osimertinib and apatinib as third-line treatment had disease progression, and genetic testing suggested the coexistence of EGFR L718V and EGFR L858R mutations. The patient chose afatinib and apatinib as fourth-line treatment, with a PFS of more than 18 months [55]. One study also included a total of 14 lung cancer patients with the EGFR L718Q/V mutation following osimertinib therapy, 7 of whom received afatinib therapy. Patients receiving afatinib experienced a DCR of 85.7% and an ORR of 42.9% [56]. This underlines once more how well afatinib works for those with lung cancer who have established osimertinib resistance because of the L718Q/V mutation.

In one study, a total of 52 EGFR G724S-positive patients were enrolled, and 23 of them provided reliable clinical outcome data. The PFS was better in the afatinib group (n = 5, 6.2 months) than in the non-afatinib group (n = 8, 1.0 months, *p* = 0.005) and the alternative EGFR-TKIS group (n = 5, 1.8 months, *p* = 0.033) among the 13 patients treated with osimertinib who progressed [57]. In addition, some cases of the G724S resistance gene have also been reported. A patient with multi-line treatment of stage IV lung adenocarcinoma who developed the EGFR G724S resistance mutation after receiving osimertinib was repeatedly evaluated as being in PR after receiving afatinib, and the PFS was more than 9 months [58]. Another patient who had the EGFR G724S resistance mutation had a PR assessed after receiving afatinib, and the patient's PFS was 3.8 months at the time the case report was published [59]. These studies and reports suggest that afatinib is effective for lung cancer patients with the G724S mutation after resistance to osimertinib.

**4. Conclusions**

Some of the rare oncogene-dependent NSCLC subgroups, such as RET (rearrangement during transfection) or ROS1 (c-ros oncogene 1) rearrangements, or BRAF (v-Raf mouse sarcoma virus oncogene homolog B) mutations, share a frequency range with rare EGFR mutations, such as exon 18 G719X or exon 20 S768I [2]. We ought to discover more about

these mutations and be capable of figuring out the corresponding treatment options for these mutations. Afatinib's use has decreased dramatically in recent years because of the creation of third-generation EGFR-TKIs, although it is still essential for the treatment of uncommon EGFR mutations.

The results in this article support the notion that afatinib has an overall advantage over first-generation EGFR-TKIs in the treatment of rare EGFR mutations and is adequate as a first-line treatment preference for NSCLC patients with major uncommon and compound mutation categories among rare EGFR mutations. It is a viable alternative for those patients following later-line therapy because afatinib is also efficacious for G724S or L718Q/V mutations that appear after osimertinib resistance. In addition, afatinib is also effective for patients carrying E709X and E709_T710delinsX mutations, some of the 20 exon insertional mutations, and some of the rare EGFR exon 19 mutations, and it is the preferred first-line therapy among individuals in this category. There are other patients with rare mutations who do not respond to afatinib and first-generation EGFR-TKIs. These rare EGFR mutations (e.g., 20 insertional mutations, 19 rare mutations) are highly heterogeneous and require further clarification of the patient-specific mutation subtype to select the appropriate therapy. Patients carrying rare EGFR mutations need to be provided with the appropriate basic database in the future to help them create precise and efficient personalized treatments.

The EGFR T790M mutation is considered to be the main mechanism of acquired resistance to afatinib in patients with classical EGFR mutations [42,60], but the cause of resistance in individuals with uncommon EGFR mutations remains unclear. According to one study, patients with uncommon EGFR mutations are more likely to have MET amplification than T790M mutation after afatinib resistance. In patients with uncommon mutations, the incidence of acquired EGFR T790M mutations was significantly lower than that of 19del and L858R mutations [61]. Another study with a small sample size suggests that EGFR T790M (11%) and FGFR1 amplification (11%), and MET amplification (11%), followed by CDK4 amplification (7%), PI3KCA activation (7%), RET fusion (4%), and BRAF mutation (4%) may be the main mechanisms of resistance in EGFR G719X/L861Q/S768I patients treated with afatinib [62]. Another cohort study from China investigated the mechanisms of resistance to afatinib in NSCLC patients with rare EGFR mutations. The results showed that 11.8% of patients had acquired EGFR T790M mutations, and the top three off-target mutations were TP53 (58.8%), LRP1B (17.6%), and CHD8 (17.6%). Consistent with other studies, this study also demonstrated that the rate of secondary T790M mutation in patients with rare EGFR mutations was significantly lower than that in patients with classical EGFR mutations. Intracranial progression occurred in 50% of patients treated with afatinib in this study, and subsequent treatment of progressive patients was mostly combined with antiangiogenic drugs and/or brain radiotherapy on top of afatinib [63]. Due to the large differences between different studies, the mechanisms of acquired resistance to rare mutations are not very clear, and the treatment options after resistance are also different. Further large and multi-center studies are necessary.

This article's content, which concentrates more on the medicine afatinib, has certain restrictions. In addition, chemotherapy, third-generation targeted drugs, and immunologic drugs have also been the subject of some clinical studies for the treatment of uncommon EGFR mutations. Osimertinib, a third-generation medication, has already been shown to be beneficial for treating uncommon EGFR mutations in several studies [64,65]. For patients with a single uncommon EGFR mutation, there are studies that point to first-line chemotherapy having a better OS than EGFR-TKIs (which include the first-generation of EGFR-TKIs and afatinib) [66]. Another study from China showed that in first-line treatment of patients with advanced lung adenocarcinoma with uncommon EGFR mutations, the PFS was better with first-generation EGFR-TKIs than with platinum-based chemotherapy, but the OS was inferior to chemotherapy [67]. Another study has shown that immunotherapy may be an effective treatment for patients with rare EGFR mutations in NSCLC [68]. In summary, there are no standard treatment options for patients with rare EGFR mutations,

and further studies are needed to help determine treatment options for patients with NSCLC with uncommon EGFR mutations.

**Author Contributions:** Conceptualization, Y.J. and K.L.; methodology, X.F.; software, Y.X.; validation, Y.J. and X.F.; formal analysis, Y.X.; resources, K.L.; data curation, J.L. and T.F.; writing—original draft preparation, Y.J.; writing—review and editing, Y.J. and X.F.; visualization, J.L.; supervision, T.F.; project administration, K.L. All authors have read and agreed to the published version of the manuscript.

**Funding:** This study was supported by the National Natural Science Foundation of China (82172708, 81902327).

**Data Availability Statement:** The datasets analyzed during the current study are publicly available.

**Conflicts of Interest:** The authors declare that the research was conducted in the absence of any commercial or financial relationships that could be construed as potential conflicts of interest.

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
