# Peer review of "Afatinib for the Treatment of NSCLC with Uncommon EGFR Mutations: A Narrative Review"

_curroncol, doi:10.3390/curroncol30060405_

Round 1

Reviewer 1 Report

Interesting review. Some comments that need some modifications in the text.

Afatinib is currently not the preferred option for patients with EGFR mutations. This should be corrected throughout the abstract and manuscript.

The tense of the document mixed present tense and past tense. I would recommend that the authors correct the document to homogenize the document in the present tense.

First paragraph page 2:Our results highlight the inferior re- sponse of first-generation EGFR-TKIs or chemotherapy in NSCLC patients with uncom- mon EGFR mutations versus those with common mutations. (Our results or rather these results? Please correct)

Page 3: correct this sentence, does not make sense :exon 20 insertion (median TTF = 4.2 months), T790M (median TTF = 4.7 months), and more infrequent uncommon mutations (median TTF = 4.5 months; 95% CI: 2.99.7) with longer TTF 

Page 3 : The assumption of better survival in first line is obviously biased. The wording should be revised: Moreover, first-line therapy with second- or third-generation EGFR-TKIs signifi- cantly improved patient survival relative to second- or third-line therapy (median PFS: 15.1 vs. 4.1 months, = 0.199)

Page 3 : What is second generation sequencing? Typo I assume?

Page 4 : use of the prefix "p." to present the protein variant of a mutation is not homogeneous throughout the text. I suggest removing this to have homogeneity in the document.

Page 5 : What is QRR? Typo?

Page 5 : Correct this sentence.Does not sound right : Afatinib was utilized to heal nine patients with NSCLC who'd already EGFR 20 insertional mutations in a large retrospective study, 

Page 6 : I think that the use of afatinib in exon 20 insertions is oversold in this section. From what I can see, use of afatinib in exon 20 is often deceiving and better alternatives (such as amivantamab) should be considered.  I recommend modification in this direction. 

Page 8 :The results in this article support the notion that afatinib has an overall advantage over chemotherapy. This statement is not supported by the data presented. The manuscript described a potential advantage compared to 1st gen EGFR TKIs but not compared to chemo or osimertinib. This one is also false: there is no conclusive evidence regarding the efficacy of three-generation EGFR-TKIs compared 

https://doi.org/10.1016/j.annonc.2022.02.225 https://doi.org/10.1016/j.jtho.2022.10.004.

In conclusion. This article is a great collection of information on afatinib but appears greatly biased towards this drug. Many recent papers suggests similar efficacy of osimertinib for these mutations (UNICORN registry for example). At the very least, this should be discussed in the abstract, discussion and conclusion. It is also hard for me to decipher which patients/mutations would be better addressed with chemo than afatinib, remembering that afatinib did not greatly (if at all) increased OS compared with chemo in the common mutation subgroup. 

Submitted to authors. Minor changes proposed.

Reviewer 2 Report

The authors wrote a review to summarize the efficacy of afatinib in treating NSCLC with uncommon EGFR mutations. This manuscript is well-written and organized, but I still have some concerns. 

1. The efficacy of afatinib had been explored and shown in many clinical studies (prospective or real-world data), so there is no advances in the using afatinib. This manuscript is a narrative review. 

2. For the anti-tumor mechanism afatinib and EGFR mutations, it is better to present this by using a figure legend.  

Reviewer 3 Report

It seems that afatinib is very effective in uncommon mutations, what is the drug application in case of concurrent ROS1 and EGFR mutations?

Please point out that the purpose of the review is to provide a comprehensive oveview of the drug activity. Provide additional information on the methods and databases through which the literature search was conducted

Please explain in the text the meaning of QRR and TTF. We need to focus and bring out what mutation is most sensitive to afatinib and what are the mechanisms of drug resistance and next steps . Pleae highlight the mutation frequency of different exons mutation. In the discussion the therapeutic options in case of TKIs resistances and disease progression should be mentioned

I suggest to include the following references useful for discussion

-Future Oncol. 2018 Jun;14(13s):29-40. 

-Biomed Res Int. 2018 Aug 27;2018:6278403.

Round 2

Reviewer 1 Report

I am happy to the changes provided by the authors. I approve this version

Reviewer 2 Report

I think the authors did respond my comments, and I don't have further criticism for it.